# Difficulty in chirality recognition for Transformer architectures learning chemical structures from string representations

Yasuhiro Yoshikai[1,2], Tadahaya Mizuno[1,2] ✉, Shumpei Nemoto[1] & Hiroyuki Kusuhara[1]

Recent years have seen rapid development of descriptor generation based on representation learning of extremely diverse molecules, especially those that apply natural language processing (NLP) models to SMILES, a literal representation of molecular structure. However, little research has been done on how these models understand chemical structure. To address this black box, we investigated the relationship between the learning progress of SMILES and chemical structure using a representative NLP model, the Transformer. We show that while the Transformer learns partial structures of molecules quickly, it requires extended training to understand overall structures. Consistently, the accuracy of molecular property predictions using descriptors generated from models at different learning steps was similar from the beginning to the end of training. Furthermore, we found that the Transformer requires particularly long training to learn chirality and sometimes stagnates with low performance due to misunderstanding of enantiomers. These findings are expected to deepen the understanding of NLP models in chemistry.

Recent advancements in machine learning have influenced various studies in chemistry such as molecular property prediction, energy calculation, and structure generation[1–6]. To utilize machine learning methods in chemistry, we first need to make computers recognize chemical structures. One of the most popular approaches is to use chemical language models, which are natural language processing (NLP) models fed with strings representing chemical structures such as simplified molecular input line entry specification (SMILES)[7]. In 2016, Gómez-Bombarelli et al. applied a chemical language model using a neural network for descriptor generation and created a trend[8–10]. In this approach, a neural NLP model such as a recurrent neural network (RNN) learns an extremely wide variety of SMILES from public databases[11–13], converts the string into a low-dimensional vector, decodes it back to the original SMILES, and then the intermediate vector is drawn out as a descriptor. The obtained descriptor is superior to the conventional fingerprints, such as MACCS keys[14] and ECFP[15], in continuous and thus highly expressive natures, and that the original

structures can be restored from the descriptor by the decoder[16]. On the other hand, the presented approach also has the disadvantage that it obscures the process of descriptor generation and that the meanings of each value in the descriptor are hard to interpret. It is scarcely studied how chemical language models understand structures of extremely diverse molecules and connect chemical structures and descriptors. Related works are described in Supplementary Note 1.

In this study, we tackle with addressing this black box by comparing the performance of the model and its descriptor at various steps of training, which clarifies what types of molecular features are easily incorporated into the descriptor and what types are not. Particularly, we focus on the most prevalent NLP model, the Transformer, a well utilized architecture for descriptor generation and other chemical language tasks these days[17–32]. To be specific, we train a Transformer model to translate SMILES strings and then compare perfect agreement and similarity of molecular fingerprints between prediction and target at different training steps. We also conduct 6 molecular

[1]Laboratory of Molecular Pharmacokinetics, Graduate School of Pharmaceutical Sciences, The University of Tokyo, 7-3-1 Hongo, Bunkyo, Tokyo, Japan. [2]These authors contributed equally: Yasuhiro Yoshikai, Tadahaya Mizuno. ✉e-mail: tadahaya@gmail.com

property prediction tasks with descriptors generated by models at different steps in training and studied what kinds of tasks are easily solved. We further find that the translation accuracy of the Transformer sometimes stagnates at a low level for a while and then suddenly surges. To clarify the cause of this, we compare translation accuracy for each character of SMILES. Finally, we search for and found methods to prevent stagnation and stabilize learning.

## Results

### Partial/overall structure recognition of the Transformer in learning progress

To understand how the Transformer model learns the diverse chemical structures, we first researched the relationship between the learning procedure and the model performance by comparing the models at various training steps. In this study, we trained the Transformer to predict canonical SMILES of molecules based on their randomized SMILES[32–34]. For models at various steps of training, we calculated *perfect accuracy* and *partial accuracy* of predicted SMILES expression[35]. We supposed *perfect accuracy*, which evaluates the complete consistency of target and prediction, represents how much the models understand the connectivity of atoms constituting overall molecular structures, whereas partial accuracy, which measures position-wise accuracy of prediction, indicates recognition of the connectivity of atoms in partial structures. The result showed that partial accuracy rapidly converged to 1.0, meaning almost complete translation, whereas perfect accuracy gradually increased as the learning proceeded (Fig. 1a). This result suggests that the Transformer model recognizes partial structures of molecules at quite an early

stage of training when overall structures are yet to be understood well. To further evaluate partial and overall recognition of molecules, we prepared the models when perfect accuracy surpassed 0.2, 0.5, 0.7, 0.9, 0.95, and 0.98 and at steps 0, 4000, and 80,000 (end of training). For models at these steps, we computed MACCS keys[14] and ECFP[15] (radius $R$ = 1, 2, 3) of predicted/target molecules; and calculated the Tanimoto similarity for each prepared model. As these descriptors are widely accepted to represent typical partial structures of molecules, their similarity between target and prediction will help the comprehension of the model on partial structures of molecules. As a result, the Tanimoto similarity of molecular fingerprints saturated at nearly 1.0, meaning complete correspondence of fingerprint between prediction and target, when perfect accuracy was merely about 0.3 (Fig 1a, b). We also compared the Tanimoto similarity with the loss function (Fig. 1b), and it was shown that the fingerprints corresponded almost completely when the loss had yet to be converged. Fig. 1c shows valid examples of predicted molecules with their targets at early phase of training (step 4000). These results also support the early recognition of partial structures and late recognition of the overall structure of molecules by the Transformer model. We previously found that the GRU model, derived from NLP, has a similar tendency as this finding[35]. It is then suggested that NLP models, when trained to chemical structures by learning SMILES, recognize partial structures of molecules at the early stage of training, regardless of their architecture. These findings can be explained as follows: When translating randomized SMILES into canonical SMILES, the model needs to reproduce the numbers of atoms in the molecule with each atomic type and how they are bound together. Assuming that the numbers of atoms are

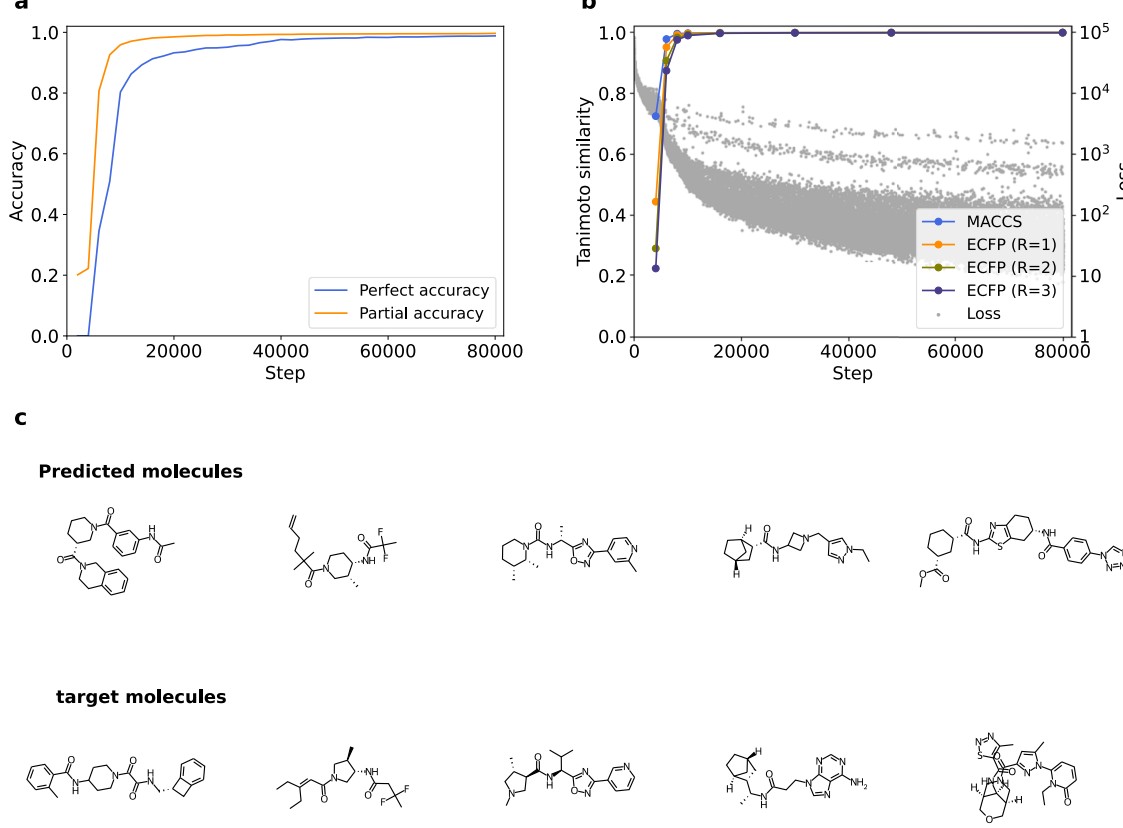

**Fig. 1 | Partial/overall structure recognition of Transformer in learning progress. a** Temporal change of perfect accuracy and partial accuracy. **b** Temporal change of Tanimoto similarity between the indicated fingerprints of predicted and target Simplified Molecular Input Line Entry System (SMILES), with the loss for comparison. Each gray dot indicates the loss of each batch. **c** Examples of predicted/target molecules at step 4000. Valid SMILES was filtered from SMILES in the test set predicted at step 4000, and the figure shows the randomly sampled examples of valid SMILES. Each of the molecules in the upper row is predicted targeted to the directly below molecule. Source data are provided as a Source Data file.

easily learned as simple frequency distributions, a large part of the training is spent learning and reproducing the bonds between atoms. The model in the early phase of training can only reproduce easy-to-understand bonds between atoms (such as those with high frequency and few other options), and those partially connected atoms are observed as partial structure. The model in the subsequent phase of training learns the remaining bonds which are left to be challenging tasks for the model but necessary for reconstructing the overall structures.

We also investigated what type of substructure is easily or hardly understood by the model using dimension-wise similarities of MACCS keys, whose details are written in Supplementary Note 2. No remarkable tendency was observed in the results, and the similarities of most of the dimensions converged rapidly.

### Downstream task performance in the learning progress

Molecular descriptors are frequently used in solving cheminformatics tasks. Therefore, in many cases, the performance of descriptor generation methods is evaluated by how much downstream tasks, such as prediction of molecular properties, are solved from their descriptor. On the other hand, we have shown in a previous study that in the case of a descriptor generated by chemical language model based on GRU, downstream task performance is mainly related to the recognition of partial structures of molecules[35], and here we worked on the evaluation of the downstream task performance over the learning progress about the Transformer model. To be specific, we predicted the molecular properties from intermediate representation of molecules during translation in the Transformer at different steps. The details of descriptor generation and conditions of prediction are described in Molecular property prediction Section. Note that to evaluate the performance of memory expression itself, rather than the inherent architecture of the model, we did not conduct fine-tuning. We used benchmark datasets from MoleculeNet[36] summarized in Table 1.

Figure 2 and Supplementary Figs. 1 and 2 show the prediction scores of each descriptor (also summarized in Table 2). The results showed that descriptors of models at an early phase, or even at the beginning of training, can perform just as well as that of the fully trained model, except for the prediction of Lipophilicity, although the score for this task saturated at an early phase (step 6000). Duvenaud et al.[37] showed that neural fingerprint (NFP), a deep-learning and graph-based descriptor, correlated to ECFP and was able to predict molecular properties without training. Similarly, one of the explanations of the presented result is that the Transformer model, even with its initial weights, generates a meaningful descriptor by its inherent mechanism such as self-attention. This implies that the modifying structure of the model is more helpful for improving the performance than changing what data the model is fine-tuned on. Note that the performance of the descriptor pooled from the Transformer memory is almost similar to that of ECFP and is slightly lower than that of CDDD. One of explanations of slightly low performance can be that the pooling process in descriptor generation omitted part of the structural information of

molecules which is scattered in the whole memory. The potential for further structural enhancements in prediction, like co-learning of molecular properties, is worth noting although it falls outside the scope of this current study.

### Stagnation of perfect accuracy in learning chemical structures

We experimented with different random seeds to reproduce the results in Partial/overall structure recognition of the Transformer in learning progress. It was then observed that the perfect accuracy of the Transformer sometimes stagnated at a low level for a while and then abruptly increased at a certain step. We are interested in this phenomenon and conducted some experiments changing the randomly determined conditions. To be specific, we trained the model with 14 different initial weights and 2 different orders of iteration on the training dataset. Figure 3a shows the perfect accuracy in these different conditions. The figure shows that while perfect accuracy uneventfully converges in many cases perfect accuracy sometimes stayed at ~0.6 from approximately 10,000 to 70,000 steps and then surged to nearly 1.0 or even maintained a low accuracy until the end of training. Figure 3b shows change of loss in conditions in which stagnation did or did not occur. This shows that the loss sharply decreased at the same time as accuracy surged.

To specify the determinative factor of the stagnation, we obtained the steps when accuracy exceeded 0.7 and 0.95, named *step-0.7* and *step-0.95* respectively. Based on Fig. 3a, we considered *step-0.7* to represent the step when stagnation was resolved, and *step-0.95* is the step when learning was almost completed. Supplementary Fig. 3a, b shows the relationship between *step-0.7/0.95* of the same seed and different iteration orders. The result shows that the trend of learning progress is similar for different iteration orders when the same initial weight was used. Supplementary Fig. 3c, d shows the average *step-0.7/0.95* of each iteration order, and no significant difference of *step-0.7/0.95* was observed. These results suggest that whether the stagnation occurs or not depends on initial weight, rather than iteration order.

We replicated the experiments in Sections Partial/overall structure recognition of the Transformer in learning progress and Downstream task performance in the learning progress for one training in which stagnation occurred. We investigated the agreement of fingerprints and performance on downstream tasks at different steps of the learning with stagnation. As a result, the tendencies in found in the previous sections was conserved even when stagnation occurred, reinforcing our findings in the previous sections. Details about these experiments are shown in Supplementary Note 3.

### Cause of stagnation in learning chemical structures

What is the cause of this stagnation? We investigated the model performance on each character of SMILES strings using 2 metrics. The first one is the perfect accuracy when each character is masked. This is calculated like perfect accuracy defined in Partial/overall structure recognition of the Transformer in learning progress Section except

### Table 1 | Summary of datasets for downstream tasks

| Dataset | | # Total molecules | # Used molecues | Splitting | Task type | Recommended metric |
|---|---|---|---|---|---|---|
| | task | | | | | |
| ESOL | | 1128 | 1108 | Scaffold | Regression | RMSE |
| FreeSolv | | 642 | 628 | Random | Regression | RMSE |
| Lipophilicity | | 4200 | 4190 | Scaffold | Regression | RMSE |
| BACE | | 1513 | 1471 | Scaffold | Classification | ROC-AUC |
| BBBP | | 2050 | 1899 | Scaffold | Classification | ROC-AUC |
| ClinTox | CT_TOX | 1484 | 1341 | Random | Classification | ROC-AUC |
| | FDA_APPROVED | 1484 | 1341 | | Classification | ROC-AUC |

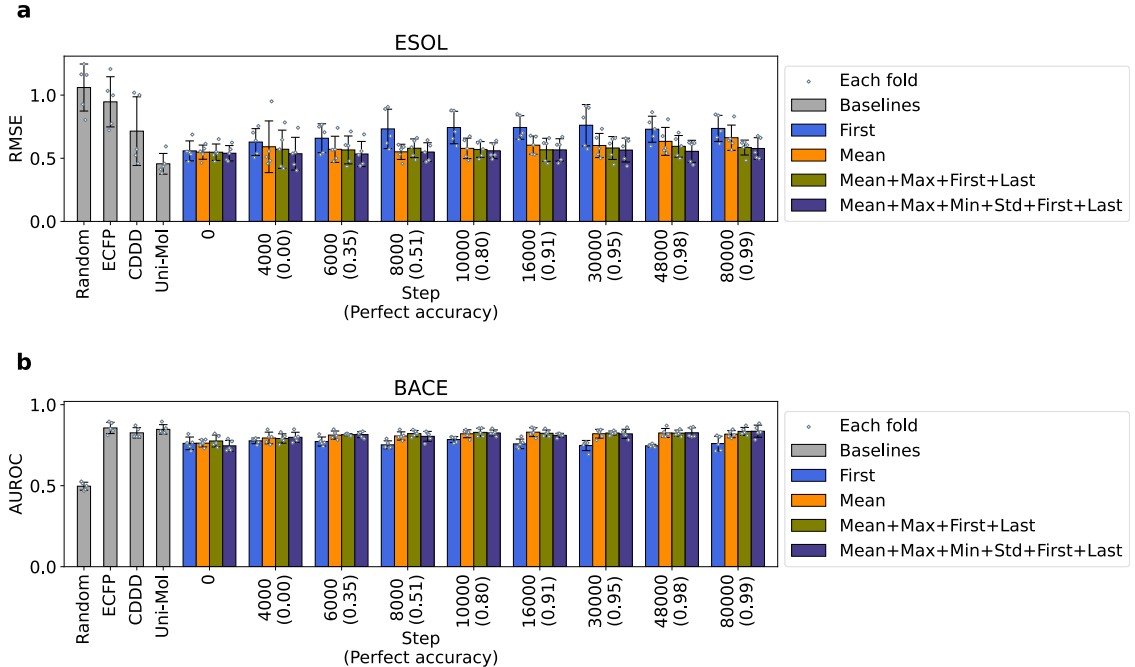

**Fig. 2 | Performance of descriptors on molecular property prediction. a** RMSE score of prediction on ESOL dataset from descriptors of the model at different steps of training, for 4 different ways of pooling. Blue, mean; yellow, latent representation of the first token; red, concatenation of the indicated 4 aggregation methods; navy, concatenation of the indicated 6 aggregation methods. **b** AUROC score of prediction on BACE dataset from descriptors of the model at different steps of training for 4 different ways of pooling. Mean, unbiased standard deviation and data distribution of experiments for 5 folds split by recommended method in DeepChem[50] are shown as bar height, error bar length and gray dots, respectively. The metrics were determined based on MoleculeNet[36]. The perfect accuracy at each step is written down on the horizontal axis. Source data are provided as a Source Data file. RMSE Root Mean Squared Error, ESOL Estimated Solubility, AUROC Area Under Receiver Operating Characteristic.

**Table 2 | Performance of each descriptor on molecular property prediction (Summary)**

| Descriptor | Steps | ESOL (RMSE) | FreeSolv (RMSE) | Lipophilicity (RMSE) | BACE (AUROC) | BBBP (AUROC) | ClinTox CT_TOX (AUROC) | FDA_APPROVED (AUROC) |
|---|---|---|---|---|---|---|---|---|
| random | | 1.060 ± 0.186 | 1.023 ± 0.070 | 1.002 ± 0.026 | 0.497 ± 0.025 | 0.482 ± 0.040 | 0.475 ± 0.038 | 0.467 ± 0.098 |
| ECFP(R = 2) | | 0.947 ± 0.199 | 0.463 ± 0.065 | 0.749 ± 0.053 | **0.856 ± 0.035** | 0.852 ± 0.035 | 0.875 ± 0.041 | 0.834 ± 0.103 |
| CDDD | | 0.715 ± 0.272 | 0.320 ± 0.032 | 0.677 ± 0.038 | 0.826 ± 0.032 | 0.874 ± 0.054 | 0.895 ± 0.016 | 0.882 ± 0.041 |
| Uni-Mol | | **0.456 ± 0.082** | **0.295 ± 0.032** | **0.505 ± 0.053** | 0.847 ± 0.029 | 0.861 ± 0.042 | 0.874 ± 0.048 | 0.875 ± 0.053 |
| Transformer | 0 | 0.548 ± 0.065 | 0.485 ± 0.053 | 0.897 ± 0.022 | 0.776 ± 0.037 | 0.845 ± 0.067 | 0.859 ± 0.045 | 0.780 ± 0.047 |
| | 4000 | 0.571 ± 0.151 | 0.398 ± 0.029 | 0.821 ± 0.041 | 0.791 ± 0.029 | 0.862 ± 0.057 | 0.899 ± 0.032 | 0.883 ± 0.051 |
| | 6000 | 0.566 ± 0.111 | 0.424 ± 0.051 | 0.775 ± 0.030 | 0.815 ± 0.006 | **0.881 ± 0.032** | 0.869 ± 0.035 | 0.862 ± 0.033 |
| | 8000 | 0.579 ± 0.074 | 0.464 ± 0.068 | 0.774 ± 0.034 | 0.821 ± 0.019 | 0.875 ± 0.045 | 0.871 ± 0.042 | 0.868 ± 0.066 |
| | 10000 | 0.570 ± 0.063 | 0.459 ± 0.065 | 0.782 ± 0.034 | 0.829 ± 0.025 | 0.859 ± 0.064 | 0.837 ± 0.062 | 0.819 ± 0.065 |
| | 16000 | 0.567 ± 0.091 | 0.486 ± 0.064 | 0.804 ± 0.033 | 0.823 ± 0.020 | 0.876 ± 0.047 | 0.876 ± 0.023 | 0.804 ± 0.088 |
| | 30000 | 0.581 ± 0.091 | 0.497 ± 0.104 | 0.793 ± 0.024 | 0.825 ± 0.012 | 0.863 ± 0.082 | 0.845 ± 0.032 | 0.799 ± 0.049 |
| | 48000 | 0.594 ± 0.086 | 0.496 ± 0.047 | 0.801 ± 0.033 | 0.824 ± 0.019 | 0.875 ± 0.043 | 0.888 ± 0.025 | 0.850 ± 0.049 |
| | 80000 | 0.585 ± 0.059 | 0.461 ± 0.058 | 0.771 ± 0.024 | 0.835 ± 0.024 | 0.861 ± 0.062 | **0.904 ± 0.024** | **0.894 ± 0.046** |

Bold figures are the best score for each dataset among the models.

that prediction for a certain type of characters in the target is not considered. This value is expected to rise when a difficult, or commonly mistaken character is masked. The second metric is the accuracy of each character of the target when teacher forcing is used. In the test phase, as the model usually predicts each letter of SMILES from a previously predicted string, the model is likely to make an additional mistake when it has already made one. This means characters that appear more in the later positions (like ")" compared with "(") tend to show low accuracy. To remedy this, we adopted teacher-forcing[38] to predict the SMILES, meaning the model always predicts each letter

from the correct SMILES string, when computing the accuracy of each character.

Figure 4a shows the transition of masked accuracy about training with or without stagnation. The results show that when the model is in stagnation, predictions of "@" and "@@" are wrong by a large number. These 2 characters are used to describe chirality in SMILES representation (Fig. 4b). It suggests that stagnation was caused by confusion in discriminating enantiomers, and the subsequent surge of perfect accuracy was the result of the resolution of this confusion. We also investigated the ratio of correct predictions, wrong predictions due

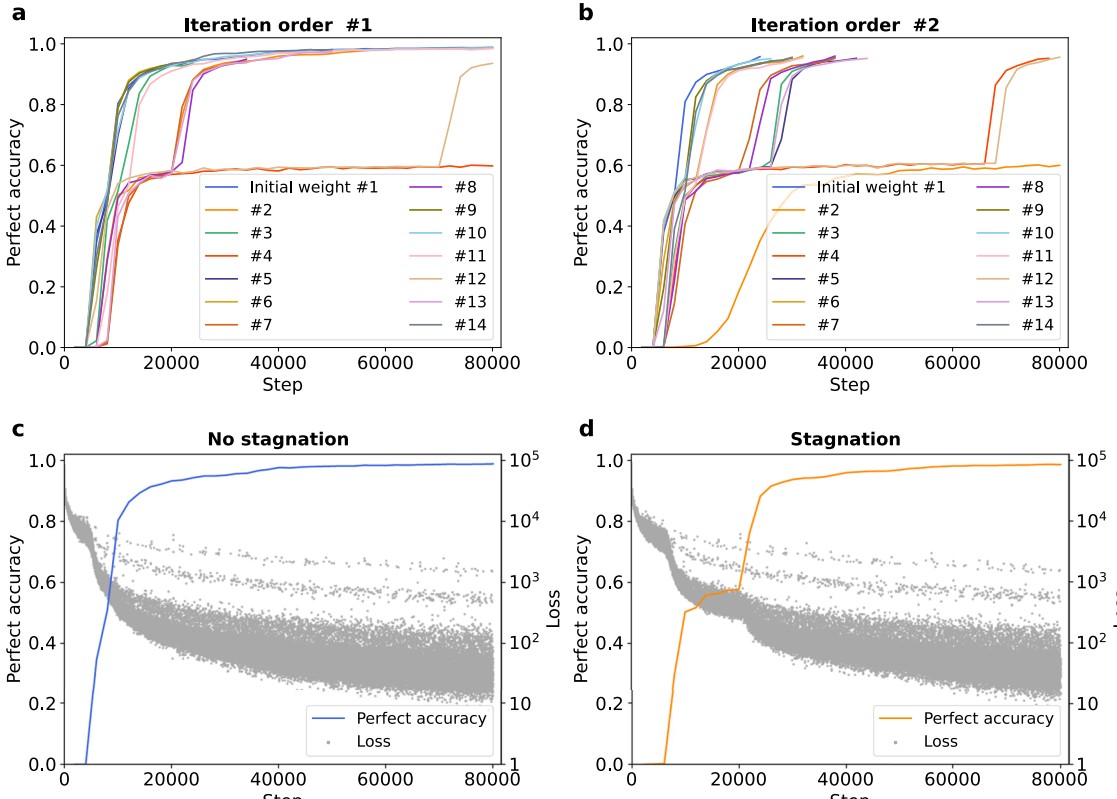

**Fig. 3 | Stagnation of perfect accuracy at different initial weights. a, b** Temporal change of perfect accuracy for 14 different seeds and 2 different iteration orders. Lines with the same color corresponds to trainings from the same initial weight. **c, d** Perfect accuracy in the trainings with/without stagnation compared to loss function. Each gray dot indicates the loss of each batch. Source data are provided as a Source Data file.

solely to chirality, and wrong predictions due to other reasons. Figure 4c shows that most of the mistakes in stagnation are due to chirality, which supports the impact of chirality confusion in stagnation.

It should be noted that errors occurred in both directions, involving mistakes of the "@" token for "@@", and vice versa (Supplementary Fig. 4a). Additionally, we conducted an analysis of the accumulated count of "@" and "@@" tokens in the training set. As depicted in Supplementary Fig. 4b, the occurrence of the "@@" token surpasses that of the "@" token in the target SMILES (canonical SMILES), while the difference appears to be marginal. In light of this observation, it is worth noting that stagnation did occur in some cases even when we took measures to train the model on a dataset that was well-balanced in terms of "@" and "@@" tokens (Supplementary Note 4). These findings underscore that bias regarding chiral token is not the primary cause of stagnation.

The true challenge for a chemical language model lies not in mastering the numerous elementary atom-bonding rules, but rather in comprehending the difficult rules that persist even after the basics have been acquired, which is consistent with Ucak's work regarding the reconstruction of molecular representations from fingerprints[39]. The above findings indicate that learning chirality, which is the cause of stagnation, is the very difficult task for the Transformer model learning diverse chemical structures. Notably, the examination of the accuracy transition for each character not only confirmed that "@" and "@@" show a slower increase of accuracy than the other characters, but also revealed the difficulty of chiral tokens even when learning proceeds smoothly without stagnation (Supplementary Fig. 5). This trend holds true considering character frequencies (Supplementary Figs. 6 and 7). In addition, the accuracy for chiral tokens is relatively low even when the model reaches a plateau, a perfect accuracy of 0.9. It should be noted that the accuracy increase of "\" token, related to geometrical

isomers, was also slow (Supplementary Fig. 5). This suggests that the tokens associated with stereochemistry pose a general challenge for the Transformer model in terms of learning, while the influence of geometrical isomers remains relatively modest, owing to their infrequent occurrence (approximately 1/100th of the chiral tokens).

These findings indicate that chirality of molecules is difficult for the Transformer to understand, and sometimes the model is confused about it for a long period, causing stagnation.

## Solution of stagnation in learning chemical structures

Then, how can we facilitate the understanding of the Transformer on chirality? To answer this question, we applied the following perturbation to the learning process and evaluated its effect on stagnation.

1.  Increase chirality in training dataset: It is possible that learning more enantiomers encourages the model to understand chirality. We therefore omitted half of the molecules in the training set whose SMILES has neither "@" nor "@@" and trained the model with the data in which chirality appears more frequently.
2.  Introduce AdamW: In deep-learning studies, one of the possible explanations for this kind of stagnation is that the model is stuck to a local optimum, and changing the optimizer can be a solution to avoid such stagnation. We have been using the Adam optimizer based on Vaswani et al.[31] so far, but here we tried the AdamW[40] optimizer. The AdamW optimizer is a refined optimizer of Adam with $L^2$ normalization in the loss function. Loshchilov et al.[40] showed that this optimizer can be adopted to a wider range of learnings than Adam.
3.  He normal initialization: Experiments in 4.3 suggested that stagnation occurs depending on the initial weight of models. Thus, changing the initialization of model weight would stabilize learning. Here we introduced He normal initialization, which is

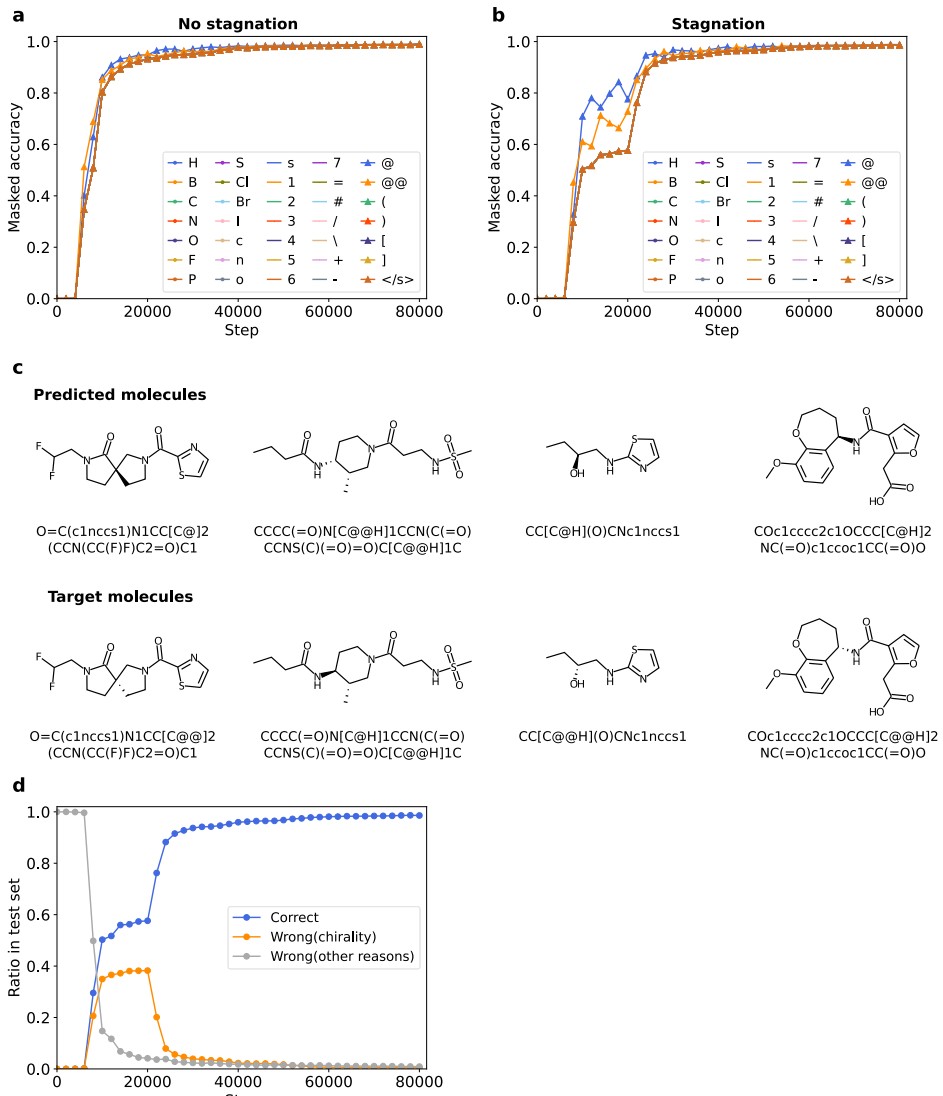

**Fig. 4 | Difficulty in learning chirality for Transformer. a, b** Temporal change of perfect accuracy when each one of the characters in Simplified Molecular Input Line Entry System (SMILES) was masked for trainings in which stagnation did/did not occur. Rare tokens which did not appear in the test set are not shown. **c** Examples of target and predicted molecules during stagnation (at step 10,000).

Each of the molecules in the upper row is predicted targeted to the directly below molecule. **d** Ratio of correct predictions, predictions with only mistakes of "@" token for "@@" token and "@@" token for "@" token (mistakes attributed to chirality), and predictions with other mistakes in the test set. Source data are provided as a Source Data file.

referred to as suitable for the ReLU activation function in the Transformer.

4. pre-LN structure: Pre-LN is a structural modification of the Transformer first proposed in Xiong et al.[41] to stabilize learning. This method prevents vanishing gradients in the lower layer of the Transformer by ensuring that the residual connection is not affected by layer normalization, which can cause vanishing gradients. This method has been shown to stabilize the learning of the Transformer[41].

All these perturbations were respectively introduced to the baseline model, and training was conducted 14 times with different initial weights for each modification, except for the introduction of He normal, which showed a significant delay in learning and was aborted when 5 studies were finished. Considering the computational cost, we stopped training when perfect accuracy reached 0.95.

Supplementary Fig. 8a, b shows the average of *step-0.7/0.95*. In some cases where the accuracy did not reach 0.7/0.95, *step-0.7/0.95* was defined as 80,000 (end of training). The result showed that the

introduction of pre-LN significantly accelerated the learning speed, whereas other modifications did not achieve improvement. Figure 5a shows the changes in accuracy over time in the 14 trainings with pre-LN, compared with those about the control model. This figure also demonstrates that pre-LN strongly stabilizes learning.

Then, does pre-LN facilitate understanding of chirality, or simply accelerate overall learning? Fig. 5b and Supplementary Fig. 9 show the masked accuracy and accuracy for each character in one of the studies in which pre-LN was adopted. The results show that while learning on all tokens is accelerated, "@" and "@@" is relatively slow to be learned even in the model with pre-LN, suggesting that pre-LN accelerates the learning of not only chirality but also molecular structure in general.

## Investigation with another chemical language

Finally, to clarify the generalizability of our findings about the Transformer, we trained the model with another expression of molecules. Instead of SMILES, here we used InChI, an alternative literal representation of molecules adopted in some cheminformatics studies with chemical language models. Although the performances of chemical

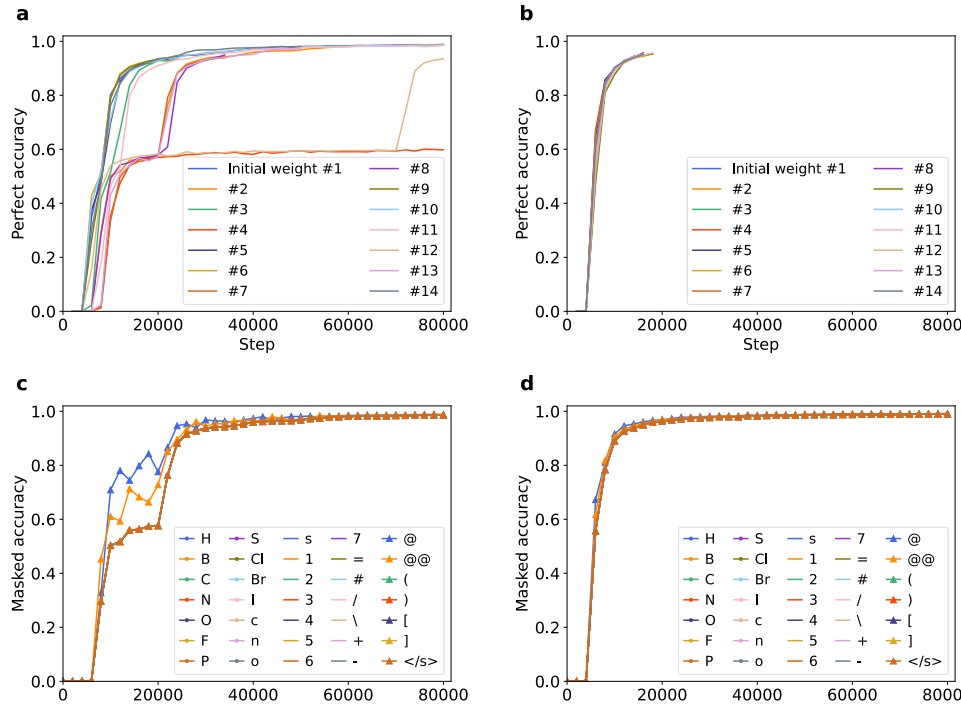

**Fig. 5 | Improvement of stagnation and recognition of chirality by the introduction of pre-LN. a, b** Temporal change of perfect accuracy started from 14 different initial weights with post/pre-layer normalization(post-LN/pre-LN) structure. Lines with the same color corresponds to training from the same initial weight. **a** is reproduced from Fig. 3a for comparison. **c, d** Temporal change of perfect accuracy when each one of the characters in Simplified Molecular Input Line Entry System (SMILES) was masked for trainings with post-LN/pre-LN structure. **c** is reproduced from Fig. 4b for comparison. Source data are provided as a Source Data file.

language models fed with InChI are reported to be inferior to those with SMILES[10,42], and therefore InChI expression is not used frequently, the translation task between InChI and SMILES guarantees that the model learns to extract essential connectivity of atoms constituting molecules because these representations are completely different, whereas randomized SMILES and canonical SMILES follows the same grammar. The details about InChI-to-SMILES translation are written in Supplementary Note 5.

The result showed early saturation of partial accuracy and fingerprint similarity compared to perfect accuracy and loss function, indicating that the recognition of partial structures is easier than overall structures in InChI-to-SMILES translation (Fig. 6a, b). The performance on downstream tasks was not improved by training (Supplementary Figs. 10 and 11). The results also revealed that the stagnation did occur in InChI-to-SMILES translation (Fig. 6a), and character-wise analysis exhibited that confusion in discriminating enantiomers caused it (Fig. 6c, d and Supplementary Fig. 12). In addition, pre-LN introduction relieved the stagnation (Fig. 6e), and character-wise accuracy showed overall acceleration of saturation and relatively low accuracy of chiral tokens when pre-LN was adopted, indicating that this structure generally accelerates understanding of chemical structures (Supplementary Fig. 13). These results suggest that what we have found about the Transformer model trained by random-to-canonical SMILES translation is an innate property of Transformer, rather than a grammatical or processive problem specific to SMILES.

## Discussion

In recent years, a new field of research has been established in which NLP models, especially the Transformer model, is applied to literal representations of molecules like SMILES to solve various tasks handling molecular structures: chemical language models with neural network[7]. In this paper, as a basic study of chemical language models for descriptor generation, we investigated how a Transformer model understands diverse chemical structures during the learning progress. We compared the agreement between the output and the target, and the fingerprints related to substructures, for the models in the process of learning. The performance of the descriptor generated by the model under training was also examined on the downstream tasks of predicting molecular properties. We further found that perfect accuracy of translation sometimes stagnates at a low level depending on the initial weight of the model. To find the cause of this phenomenon, we compared the accuracy per each character of SMILES, and we experimented with 4 alterations to prevent stagnation. The major findings in this paper are as follows:

1. In the Transformer model, partial structures of molecules are recognized in the early steps of training, whereas recognition of the overall structures requires more training. Together with our previous study about RNN models[35], this finding can be generalized for various NLP models fed with SMILES strings. Therefore, enabling the Transformer model to refer to overall structural information as an auxiliary task in its structure would help improve the descriptor generation model.

2. For molecular property prediction, the performance of the descriptor generated by the Transformer model may already have been saturated before it was trained, and it was not improved by the subsequent training. This suggests that the descriptor of the initial model already contains enough information for downstream tasks, which is perhaps the partial structures of molecules. On the other hand, it is also possible that downstream tasks like property prediction of molecules are too easy for the Transformer and inappropriate for evaluating Transformer-based descriptor generation methods[31].

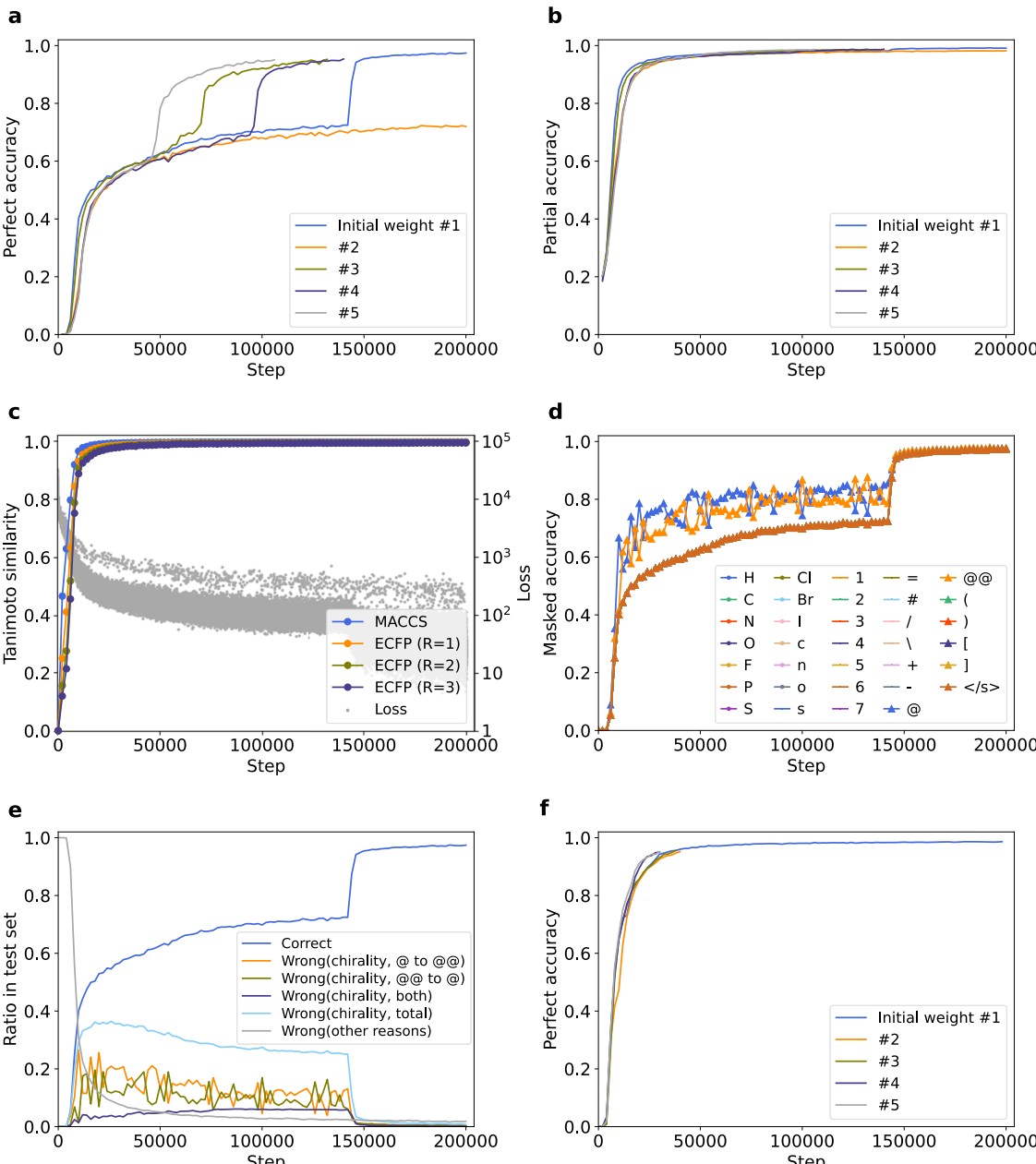

**Fig. 6 | Experiments with the Transformer model trained by InChI-to-SMILES translation. a** Temporal change of perfect accuracy of the model trained by InChI-to-SMILES translation started from 5 different initial weights. **b** Temporal change of partial accuracy of the model trained by InChI-to-SMILES translation started from 5 different initial weights. **c** Temporal change of Tanimoto similarity between the indicated fingerprints of predicted and target SMILES, with the loss for comparison. Each gray dot indicates the loss of each batch. **d** Temporal change of perfect accuracy when each one of the characters in SMILES was masked for one of the trainings. Rare tokens which did not appear in the test set are not shown. **e** Ratio of molecules in the test set which were correctly predicted, and molecules which were mistakenly predicted for each reason. Wrong predictions were classified by whether they only have "@"-to-"@@" or "@@"-to-"@" mistakes, or also have other types of mistakes. **f** Temporal change of perfect accuracy and partial accuracy started from 5 different initial weights with pre-LN structure. Initial weights were not shared between post/pre-LN here. Source data are provided as a Source Data file. InChI International Chemical Identifier, SMILES Simplified Molecular Input Line Entry System, Pre-LN pre-Layer Normalization.

3. Translation performance of the Transformer concerning chirality is relatively slow to rise compared to the other factors, such as overall structures or other partial structures, and the model is sometimes confused about chirality for a long period, causing persistent stagnation in whole structure recognition. This suggests that additional structures or tasks that teach chirality to the model can improve the performance of the model and its descriptor.

4. Introducing the pre-LN structure accelerates and stabilizes learning, including chirality.

These discoveries contribute to clarify the black box in chemical language models and are expected to activate this field. It is an intriguing future task to investigate whether these findings hold true in chemical language models for other applications with supervised natures such as structure generation and end-to-end property prediction, although we focused on descriptor generation in this study because this task makes the model purely learn chemical structures in an unsupervised manner. Chemical language models would be increasingly developed, as NLP is one of the most advanced fields in

deep learning. On the other hand, there are many unknowns in the relationship between language models and chemical structures compared to prevalent neural network models in the field of chemistry, such as graph neural networks[43,44]. Further basic research on the relationship between NLP models and chemical structures is expected to further clarify the black box about how NLP models evolve and recognize chemical structures, leading to the development and performance improvement of chemical language models for various tasks in chemistry.

## Methods

### Training a Transformer

**Dataset.** To pretrain a Transformer model, molecules were sampled from the ZINC-15 dataset[11], which contains approximately 1 billion molecules. After rough sampling of molecules from the whole dataset, Molecules with atoms other than H, B, C, N, O, F, P, S, Cl, Br, or I, and molecules that have more than 50 or less than 3 heavy atoms were filtered out, 30,190,532 (approximately 30 M) molecules were sampled for train and test set. The molecules were sampled not randomly, but in a stratified way in terms of SMILES length to reduce the bias of molecular weights and SMILES lengths in the training set, i.e., all molecules in ZINC-15 were classified according to lengths of SMILES strings, and 460,000 molecules were sampled from each length. All molecules were sampled for lengths which do not contain more than 460,000 molecules. The initial rough sampling was also stratified similarly not to omit molecules with rare length of SMILES. About 3% (9057) molecules were sampled as the test set, and the remaining molecules were used for training. Note that random sampling is widely used instead of this stratified sampling, although we believe that our sampling strategy enables fair training of molecular structures with regard to molecular weight and SMILES length. We therefore conducted the key experiments with the dataset prepared by random sampling to confirm generality and obtained similar results. The details about these experiments are written in Supplementary Note 6.

The remaining molecules were then stripped of small fragments, such as salts, and canonical SMILES and randomized SMILES of them were generated. SMILES is a one-dimensional representation of a molecule, and because any atom can be the starting point, multiple SMILES representations correspond to a single molecule. To identify one specific representation, there is a rule for selecting the initial atom, and the SMILES representation identified based on this rule is called canonical SMILES, whereas the others are referred to here as randomized SMILES. Translation between randomized and canonical SMILES is used as a training task in[32,34]. We generated randomized SMILES by renumbering all atoms in molecule[32,33]. InChI, another string representation of molecules were also generated for translation. All of these processes were conducted using the RDKit library (ver. 2022.03.02).

### Model architecture

We implemented the model with PyTorch framework (ver. 1.8, except for the model with pre-LN structure). Parameters and model architecture were determined according to the original Transformer in ref. 31; the dimension of the model was 512, the dimension of the feed-forward layer was 2048, and the number of layers in the encoder and decoder was 6. We used ReLU activation, and the dropout ratio was 0.1 in both the encoder and the decoder.

### Learning procedure

Randomized SMILES strings of molecules in the training dataset were tokenized and then fed into the encoder of the Transformer. Tokenization was conducted with the vocabulary shown in Supplementary Table 1. "<s>" and "<\s>" tokens are added to the beginning end and of the token sequences, respectively, and "<pad>" tokens are added to some of the sequences to arrange the length of them in a batch. Positional encoding was added to the embedded SMILES tokens. The input of the decoder was canonical SMILES strings of the same molecule, and the model was forced to predict the same canonical SMILES shifted by one character before, with the attention from posterior tokens being masked. Hence, the model was forced to predict each token of SMILES based on its prior tokens (teacher-forcing[38]). We calculated cross-entropy loss for each token except the padding token, and the mean loss along all tokens was used as the loss function. As for translation from InChI to canonical SMILES, the vocabulary in Supplementary Table 2 was used for tokenization.

25,000 tokens were inputted per step following[31]. Due to resource restriction, the batch size was set to 12,500 tokens, and the optimizer was stepped after every 2 batches. We introduced bucketing[10,31], that is, we classified SMILES strings in training data into several ranges of their lengths, and generated batches from SMILES in the same range of length to reduce padding. The number of batches in total amounted to 147,946 (for 73,973 steps). We used Adam optimizer[45] with a warmup scheduler (warmup step = 4000) and continued training up to 80,000 steps (slightly longer than one epoch), although training was aborted when *perfect accuracy* (described below) reached 0.95 in some studies to save time in training.

### Metrics

We measured the translation performance of each model by 2 metrics: *perfect accuracy* and *partial accuracy*[35]. *Perfect accuracy* means the ratio of molecules whose target SMILES strings were completely translated by the model, except those after end-of-string tokens (i.e., padding tokens). *Partial accuracy* means the character-wise ratio of coincidence between the target and predicted strings.

To evaluate recognition of the model about partial structures of molecules, we calculated MACCS keys and ECFP for both target and predicted SMILES and calculated the Tanimoto similarity of these fingerprints between them for the test set. The radius of ECFP was varied from 1 to 3. Because prediction of the Transformer does not always output valid SMILES, molecules for which the Transformer predicted invalid SMILES at any step were omitted when calculating Tanimoto similarity. Note that a valid SMILES means a grammatically correct SMILES which encodes a molecule that satisfies the octet rule.

We calculated the agreement of MACCS keys between the predicted and target molecules in each dimension. For each dimension, the percentage of molecules was calculated for which the model makes valid predictions and for which the predicted molecules have MACCS keys of 1, in all molecules that have MACCS keys of 1. The same percentage for bit 0 was also calculated.

### Molecular property prediction

**Dataset.** Physical and biological property data of molecules was obtained from MoleculeNet[36] with the DeepChem module[46]. Table 1 shows the datasets we used and their information. The same filtering and preprocessing were applied as in Training a Transformer Section to molecules in each dataset, although too long or short SMILES were not removed in order not to overestimate the prediction performance, and then duplicated SMILES were omitted. The model was trained and evaluated for 5 train-validation-test folds split by method recommended by DeepChem.

### Molecular property prediction task

We tested the property prediction ability for models at steps when perfect accuracy reached 0.2, 0.5, 0.7, 0.9, 0.95, and 0.98 and at steps 0, 4000, and 80,000 (end of training). In the property prediction experiments, only the encoder of the Transformer was used. Randomized SMILES were inputted into it, and then the memory

(=output of encoder) was pooled and used as the descriptor of molecules. To minimize the effect of the pooling procedure, we tested 4 pooling methods: 1) average all memory along the axis of SMILES length, 2) extract memory corresponding to the first token in SMILES, 3) obtain the average and maximum memory along the axis of the SMILES length and concatenate them with the memories of the first and last token[34], 4) concatenate average, maximum, minimum, standard deviation, beginning, and end of memory. Note that the dimensions of pooled descriptors are not equal: 1) 512, 2) 512, 3) 2048, and 4) 3072.

From these pooled descriptors, we predicted the target molecular properties with SVM, XGBoost, and MLP in our preliminary study and chose XGBoost which showed the best performance. For each of the 5 splits, we searched hyperparameters by Bayesian optimization using Optuna[47]. As baseline descriptors, we calculated ECFP ($R = 2$, dimension = 2048) and CDDD[10] (dimension = 512) for molecules in MoleculeNet datasets and measured the property prediction accuracy. Random values from a uniform distribution in [0, 1] were generated and also used as baseline descriptors (dimension = 2048). Furthermore, we compared the prediction performance with Uni-Mol[48], one of the state-of-the-art models in molecular property prediction. The experiments were conducted for 5 folds split by the methods recommended by DeepChem, and the accuracy of prediction was measured by the recommended metric by MoleculeNet.

### Experiment in different initial weight and iteration order
14 different initial weights were randomly generated with different random seeds in PyTorch, and 2 different orders of data iteration were made by randomly sampling molecules for each batch and randomly shuffling the batch iteration order with 2 different seeds. All hyperparameters were fixed during these 28 experiments in total. To perform numerous experiments, we aborted the experiments when accuracy reached 0.95 instead of continuing until step 80,000, except 4 experiments to see saturation. We calculated perfect accuracy and partial accuracy on validation set for every 2000 steps, and the step when perfect accuracy first reached 0.7/0.95 was called *step-0.7/0.95*, respectively. The mean step-0.7 and step-0.95 were compared between 2 iteration orders by two-sided Welch's t-test.

### Research for the cause of stagnation
For each character in the vocabulary, we calculated the perfect accuracy with characters of each kind masked. This means we did not check whether the characters of the selected kind were correctly predicted by the Transformer when calculating perfect accuracy. We computed partial accuracy for each character as well. Because the Transformer predicts each character from memory and previous characters it predicted, it is more likely to produce wrong predictions after it once made a mistake. We therefore adopted teacher-forcing when calculating this metric, meaning the model predicts each character with the correct preceding characters[38]. We examined the percentage of SMILES that could not be translated completely, for which the mistake was solely attributable to chirality. Since chirality of molecules is represented by "@" and "@@" tokens in SMILES, the percentage of molecules that were answered correctly except that "@" was mistaken for "@@", "@@" was mistaken for "@", or both, among all molecules were calculated.

### Structural modifications of the model to prevent stagnation
For AdamW, He normal initialization, and pre-LN (pre-layer normalization) structures, we used PyTorch implementation. As pre-LN is not implemented in PyTorch version 1.8, we conducted experiments with the pre-LN structure in version 1.10. For experiments with more "@" and "@@", training data was sampled again from the training dataset we prepared in Training a Transformer Section. SMILES strings that have either "@" or "@@" were sampled at 100% probability, and those

that do not were sampled at 50%. The new training dataset contained about 135,000 molecules (about 67,500 steps). We did not change the test set.

These modifications were introduced respectively, and the model was trained from 14 initial weights. The number of steps the model took until perfect accuracy reached 0.7 and 0.95 was compared to the control experiment with no modification by two-sided Welch's t-test with Bonferroni correction. Since we had to conduct many experiments in this section, we aborted experiments when accuracy reached 0.95 instead of continuing until step 80,000.

### Reporting summary
Further information on research design is available in the Nature Portfolio Reporting Summary linked to this article.

## Data availability
ZINC-15[11] dataset used to train and evaluate the model was downloaded from https://zinc15.docking.org/ MoleculeNet[36] dataset used to evaluate performance of molecular property prediction was downloaded through DeepChem[46] module (https://deepchem.io/). Source data are provided with this paper.

## Code availability
Used codes and trained models are available at: https://github.com/mizuno-group/ChiralityMisunderstanding[49].

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

## Acknowledgements

We thank all those who contributed to the construction of the following data sets employed in the present study such as ZINC and MoleculeNet. This work was supported by AMED under Grant Number JP 23ak0101199h0001 (T.M.) and JP22mk0101250h0001 (T.M.), and the JSPS KAKENHI Grant-in-Aid for Scientific Research (C) (grant number 21K06663, T.M.) from the Japan Society for the Promotion of Science.

## Author contributions

Yasuhiro Yoshikai: Methodology, Software, Investigation, Writing – Original Draft, Visualization. Tadahaya Mizuno: Conceptualization, Resources, Supervision, Project administration, Writing – Original Draft, Writing – Review & Editing, Funding acquisition. Shumpei Nemoto: Writing – Review & Editing. Hiroyuki Kusuhara: Writing – Review & Editing.

## Competing interests

The authors declare no competing interests.
