## [Peer Review File · Nature Communications]

Difficulty in chirality recognition for Transformer architectures learning chemical structures from string representationsReviewers' Comments:

Reviewer #1 (Remarks to the Author):

The manuscript reports the investigation of a particular type of neural network applied to SMILES strings of chemical structures, more precisely the ability of natural language processing (NLP) models at different stages of training to understand chemical structures and to derive useful molecular descriptors.

First the authors studied the recognition of chemical structures through the learning progress, as well as the performance of the derived descriptors in QSAR studies. Quality parameters were compared at different training steps. The designed methods are sound and the obtained results are relevant.

Then the authors observed that one problematic aspect is the understanding of chirality. Several experiments are described to investigate this issue (learning stagnation due to chirality) and to overcome it. This is novel to the best of my knowledge. However, the discussion of this part should be improved and more data should be provided. The following issues should be addressed in a revised version.

1. The authors mention that "stagnation was caused by confusion in discriminating enantiomers, and the subsequent surge of perfect accuracy was the result of the resolution of this confusion", but more details would be useful concerning how often the structure is wrong because the (otherwise correct) structure of the opposite enantiomer is obtained. How often do chiral molecules with fully specified chiral centers yield chiral molecules with undefined stereochemistry? How often are molecules with undefined chiral centers predicted with "@" or "@@" specified chiral centers?

2. How many pairs of enantiomers are there in the training data? Which percentage of chiral structures in the training set are represented by both enantiomers? Fig 4b shows that token "@" is better predicted than "@@", which in principle should not happen. The only difference between the canonical SMILES of opposite enantiomers is the stereolabel ("@" or "@@"). This suggests that the training data is unbalanced concerning these two tokens. In fact, this unbalance can be a cause of learning stagnation and a barrier to learn chirality. Additional experiments would be useful with a perfectly balanced training set (one in which the two enantiomers of each chiral molecule are included and in which the "@" and "@@" tokens have the same frequency).

3. Apparently (from Fig 6) training an InChI-to-SMILES model with pre-LN improves the ability to understand chirality, but the authors do not discuss it. The last sentence of section 4.6 is not clear. What do the authors mean by "this model"? The SMILES-to-SMILES NLP Transformer?

4. Point 4 of Conclusion does not explain that the pre-LN improvement in learning chirality only occurs with the InChI-to-SMILES model.

5. The title focuses on chirality, but this doesn't provide a correct idea of the whole manuscript and should be changed. The Abstract mentions chirality in just three lines, and learning chirality is presented as an additional aspect of the work, not its main theme. The same in the Conclusion. This work is essentially about how the NLP model learning of chemistry evolves during the training. The Introduction and Related Work sections don't mention chirality or stereochemistry at all. A revision of these sections is recommended, to include an overview of related works concerning deep learning applied to chiral molecules (e.g. J.Cheminform 2022, 14, 40; J.Cheminform 2019, 11, 5; arxiv 2021:2110.04383; arxiv 2020: 2012.00094; Mol. Inf. 2022, 41, 2200068).

Additional comments:

a) The code deposited at github.com should include instructions to install the software, the best final model and examples of python scripts to encode / decode SMILES strings and to derive

molecular descriptors from the Transformer memory.

b) Full specification is missing in some references, e.g. 19, 20, 30, 34, 42 and 49.

Reviewer #2 (Remarks to the Author):

The central argument of the paper posits that machines learn substructures of a molecule easier and quicker than the overall structure.

The authors attempt to substantiate this claim by incorporating fingerprint algorithms during the training phase of the SMILES-to-SMILES translation task within the Transformer framework. I have several concerns regarding methodology and design.

1. The design of the main experiment appears to be misaligned with the central claim of the authors. Once scrutinized, it becomes apparent that the translation from randomized SMILES to canonical SMILES representations converges to yield maximum accuracy swiftly following the warm-up phase. Even a modest amount of data would likely suffice to achieve precise translations, particularly with a diverse dataset like the one employed here. This is primarily because machines can readily grasp the simple rules of the SMILES algorithm, a point corroborated by Figures 1a and 1b in the paper. The true challenge for the machine, however, resides in resolving minor, yet structure-altering operations on the SMILES sequence. Critical questions arise here: Can these operations be effectively captured by the machine? Can it differentiate between them and the hundreds of other valid permutation operations? Arguably, investigating these aspects via new learning curves would provide a more insightful exploration than the current approach. Such a shift in focus could provide more information on how machines learn molecular structures. Furthermore, the challenges associated with learning chirality could also be better comprehended through this perspective (e.g., @ and @@ swaps)

2. In the context of tokenized SMILES, what the machine appears to be learning are not the substructures, which are essentially a handful of atom types, but rather the connectivities between these atom types. This arguably forms the underlying reason why predicting the entire molecular structure necessitates a longer learning duration as compared to the partial structures. However, the paper does nowhere explicitly address this aspect of learning connectivities, an omission that potentially weakens the central argument.

3. The paper exhibits an exceedingly high level of abstraction, with an unclear transition from the sequential treatment of SMILES tokens to the substructural analysis performed on MACCS and ECFP features. The training is conducted through SMILES tokens, but the authors execute a substructural analysis at both ends using entirely different substructural algorithms. This approach makes the entire analysis dependent on how substructures are defined, which could limit the generalizability of the findings. The interpretation of such a fundamental task can be effectively conducted by contemplating on the Maximum Likelihood principle.

4. The manuscript seems to be written with some oversight, containing several redundant sentences (e.g., lines 141-146) and typos (e.g., lines 79, 285)

Reviewer #3 (Remarks to the Author):

This manuscript tried to investigate the NLP model for understanding chemical structure. They employed the Transformer and investigated the status during the learning stage. Although they claimed the NLP model has difficulty for understanding the chirality, the current data is not enough to support this. The writing is hard to understand, and the logic is not very clear. More importantly, they didn't provide solution for this.

Major:

1. Although they defined two accuracies to evaluate the learning process, the transformer itself is a blackbox, and it may be determined by many factors instead of chirality. The authors need to provide a way to solve this problem for publication in this prestigious journal.
2. The authors didn't make comparison with SOTA deep learning methods for property prediction. The current embedding produced results comparable to or even worse than ECFP & CDDD.
3. Case examples are desired to deep into the structure. Currently, only overall statistics is not enough.
4. The writing needs improvement. The method section is not clear. For example, the dataset in the first processing step was 30 M, but what's the final number? It's also common to filter the original dataset and then sample the dataset. They also mentioned "conducted the key experiments with the dataset prepared by random sampling", then what's the details of the dataset. "amounted to about 150,000" needs an exact number. "aborted some of the experiments when accuracy reached..."
5. The authors need to carefully proofread/clarify the manuscript. "To the best of knowledge", "the limitation of study"; "to confirm generality and confirmed similar results"; "we calculated perfect accuracy with the selected character being masked. This means we did not check whether the selected character was correctly predicted by the", etc.

Point-by-point responses to reviewers' comments

Reviewer Comments to Author:

Reviewer: 1

The manuscript reports the investigation of a particular type of neural network applied to SMILES strings of chemical structures, more precisely the ability of natural language processing (NLP) models at different stages of training to understand chemical structures and to derive useful molecular descriptors.

First the authors studied the recognition of chemical structures through the learning process, as well as the performance of the derived descriptors in QSAR studies. Quality parameters were compared at different training steps. The designed methods are sound and the obtained results are relevant.

Then the authors observed that one problematic aspect is the understanding of chirality. Several experiments are described to investigate this issue (learning stagnation due to chirality) and to overcome it. This is novel to the best of my knowledge. However, the discussion of this part should be improved and more data should be provided. The following issues should be addressed in a revised version.

Answer

First of all, we wish to express our deep appreciation to the Reviewer for his or her insightful comments, which have helped us significantly to improve the manuscript. We have made significant revisions to our manuscript based on your comments.

Comments 1: The authors mention that “stagnation was caused by confusion in discriminating enantiomers, and the subsequent surge of perfect accuracy was the result of the resolution of this confusion”, but more details would be useful concerning how often the structure is wrong because the (otherwise correct) structure of the opposite enantiomer is obtained. How often do chiral molecules with fully specified chiral centers yield chiral molecules with undefined stereochemistry? How often are molecules with undefined chiral centers predicted with “@” or “@@” specified chiral centers?

Answer

We appreciate your helpful advice to make the argument clearer. We investigated the ratio of wrong predictions whose mistakes can be attributable solely to chirality. As a result, most of the mistakes of translation during the stagnation was found to be due solely to chirality of molecules (**Figure 4c**). Additionally, we studied how many molecules had the chiral centers increased or decreased by translation. The result showed that the number of chiral centers was rarely changed by translation while the model was in stagnation (**The figure below**). This can also be inferred from that mistakes other than chirality are rare (**Figure 4c**), and therefore this figure was not included in the manuscript.

Figure. Ratio of predictions whose chiral centers increased or decreased from target molecules.

We have added the above result and discussion to the revised manuscript as follows.

Modification

- ✓ Every wrong prediction by the model was attributed to chirality or other structures, and the ratios of each mistake was calculated (**Figure 4c**).

Comments 2: How many pairs of enantiomers are there in the training data? Which percentage of chiral structures in the training set are represented by both enantiomers? Fig 4b shows that token “@” is better predicted than “@@”, which in principle should not happen. The only difference between the canonical SMILES of opposite enantiomers is the stereolabel (“@” or “@@”). This suggests that the training data is unbalanced concerning these two tokens. In fact, this unbalance can be a cause of learning stagnation and a barrier to learn chirality. Additional experiments would be useful with a perfectly balanced training set (one in which the two enantiomers of each chiral molecule are included and in which the “@” and “@@” tokens have the same frequency).

Answer

We appreciate your insightful comment about our experiment. According to your comments, we counted the number of “@” and “@@” tokens in the randomized SMILES (input) and canonical SMILES (target) of the training set (**Supplementary Figure 4b**). There were found to be slightly more “@” tokens in the input SMILES and more “@@” tokens in the target, with no bias among steps. On the other hand, when the frequency of chiral misconfusion for both directions (either “@”-to-“@@” or “@@”-to-“@”) were counted, both of them were found to exist, indicating that the model was not making a biased prediction about these tokens (**Supplementary Figure 4a**). We increased the graphical examples of chiral mistakes in **Figure 4b** and illustrated that mistakes of both directions occurred.

Furthermore, we trained the model with training set balanced about chiral tokens with 5 different initial weights, and found that stagnation occurred in some cases. This result was appended as **Supplementary Note 4** to indicate that the bias in the training set was not the cause of difficulty in learning chirality. According to your comments, we have modified the manuscript as follows.

Modification

- ✓ The examples of decoded molecules during stagnation were increased (**Figure 4b**)
- ✓ We counted mistakes of “@” token for “@@” and “@@” token for “@” (**Supplementary Figure 4a**).

- ✓ We counted “@” and “@@” tokens in each batch and calculated the accumulated numbers of them (**Supplementary Figure 4b**).
- ✓ The model was trained with balanced training set about chiral tokens, and the transition of perfect accuracy was shown (**Supplementary Note 4 and Supplementary Figure 20**).

Comments 3: Apparently (from Fig 6) training an InChI-to-SMILES model with pre-LN improves the ability to understand chirality, but the authors do not discuss it. The last sentence of section 4.6 is not clear. What do the authors mean by “this model”? The SMILES-to-SMILES NLP Transformer?

Answer

We appreciate your notice about an important point and apologize for our vague expression. We demonstrated that pre-LN structure promotes the learning of chemical structures in general, rather than learning of chirality in particular, regarding the training of translation from random SMILES into canonical SMILES (**Figure 5 and Supplementary Figure 9**). In addition, we also observed that the effects of pre-LN is true in InChI-to-SMILES translation (**Figures 6**). However, as you pointed out, discussion about these points were vague and not enough. In order to clarify these points, we added a new experiment regarding the effect of pre-LN on InChI-to-SMILES translation and modified the manuscript. In the experiment, we measured accuracy of each character when teacher forcing was used about the InChI-to-SMILES model with or without pre-LN. As a result, while the accuracy of all tokens converged faster with pre-LN structure than without the structure, the accuracy of chiral tokens is relatively low at the early phase of training, suggesting that pre-LN generally enhances recognition of chemical structure in InChI-to-SMILES translation, as it did in random-to-canonical SMILES translation (**Supplementary Figures 12 and 13**). In addition, we clearly noted that “this model” means Transformer, regardless of how it was trained. According to your comments, we have modified the manuscript as follows.

Modification

- ✓ We added the character-wise accuracy of InChI-to-SMILES model with teacher-forcing used for various steps in training (**Supplementary Figures 12 and 13**) which illustrates relative difficulty of chiral tokens when pre-LN was or was not adopted.
- ✓ The conclusion in section 4.6 was modified (page 16-17, line 433-438).

Comments 4: Point 4 of Conclusion does not explain that the pre-LN improvement in learning chirality only occurs with the InChI-to-SMILES model.

Answer

We appreciate your thoughtful comment. As mentioned in the answer to your comments 3, pre-LN structure was found to promote understanding of chemical structure in general: both in the InChI-to-SMILES model and in the random-to-canonical SMILES model. The promoting effect of pre-LN was not specific on chirality understanding but on the understanding of all tokens.

Comments 5: The title focuses on chirality, but this doesn't provide a correct idea of the whole manuscript and should be changed. The Abstract mentions chirality in just three lines, and learning chirality is presented as an additional aspect of the work, not its main theme. The same in the Conclusion. This work is essentially about how the NLP model learning of chemistry evolves during the training. The Introduction and Related Work sections don't mention chirality or stereochemistry at all. A revision of these sections is recommended, to include an overview of related works concerning deep learning applied to chiral molecules (e.g. J.Cheminform 2022, 14, 40; J.Cheminform 2019, 11, 5; arxiv 2021:2110.04383; arxiv 2020: 2012.00094; Mol. Inf. 2022, 41, 2200068).

Answer

We appreciate your helpful comment and fully agree with you. We modified the title of this paper so

that it says this paper is dedicated to clarifying the learning process of chemical language models. We included explanation about your suggested studies on chirality in the related works, although it was difficult to mention chirality in the introduction because this paper argues the difficulty of it is something found out during the course of this research.

Modification

- ✓ Title of this paper is changed to “How does Transformer model evolve to learn diverse chemical structures?” so that it means the study on the learning process of Transformer in this paper.
- ✓ Several preceding studies on the importance of chirality were introduced as related works of this study (page 4-5, line 85-92).

Additional Comments a: The code deposited at github.com should include instructions to install the software, the best final model and examples of python scripts to encode / decode SMILES strings and to derive molecular descriptors from the Transformer memory.

Answer

We deeply apologize for insufficient preparation of codes and datasets. We enriched explanations about how to build an environment and run codes to train the model. We also prepared example dataset and codes to evaluate translation accuracy and generate descriptors. Some trained model weights were also added. According to your comments, we modified the repository as follows.

Modification

- ✓ Detailed instructions to install software, train and evaluate a model and generate descriptors were added to the github repository.
- ✓ Weights of trained models with / without stagnation and a model with pre-LN structure were stored in google drive and linked from a page in the github repository.

Additional Comments b: Full specification is missing in some references, e.g. 19, 20, 30, 34, 42 and 49.

Answer

We again apologize for insufficient preparation of the manuscript and appreciate your detailed comment. According to your comments, we modified the references you raised and checked the forms of all the other references.

Modification

- ✓ All of the references were checked and modified to contain full specifications.

Reviewer: 2

I The central argument of the paper posits that machines learn substructures of a molecule easier and quicker than the overall structure.

The authors attempt to substantiate this claim by incorporating fingerprint algorithms during the training phase of the SMILES-to-SMILES translation task within the Transformer framework. I have several concerns regarding methodology and design.

Answer

First of all, we wish to express our deep appreciation to the Reviewer for his or her insightful comments, which have helped us significantly to improve the manuscript. We have made significant revisions to our manuscript based on your comments.

Comments 1: The design of the main experiment appears to be misaligned with the central claim of the authors. Once scrutinized, it becomes apparent that the translation from randomized SMILES to canonical SMILES representations converges to yield maximum accuracy swiftly following the warm-up phase. Even a modest amount of data would likely suffice to achieve precise translations, particularly with a diverse dataset like the one employed here. This is primarily because machines can readily grasp the simple rules of the SMILES algorithm, a point corroborated by Figures 1a and 1b in the paper. The true challenge for the machine, however, resides in resolving minor, yet structure-altering operations on the SMILES sequence. Critical questions arise here: Can these operations be effectively captured by the machine? Can it differentiate between them and the hundreds of other valid permutation operations? Arguably, investigating these aspects via new learning curves would provide a more insightful exploration than the current approach. Such a shift in focus could provide more information on how machines learn molecular structures. Furthermore, the challenges associated with learning chirality could also be better comprehended through this perspective (e.g., @ and @@ swaps)

Answer

We deeply appreciate your insightful perspective. We would like to answer from two points of view, one regarding the operations we are dealing with and the other regarding the true challenge for the model.

First, regarding the former, since SMILES and InChI are different from the definition of a string, we believe that the model in this study is not learning permutation operations, but rather operations on more complex compound structures. In fact, we have added InChI-to-SMILE experiments, and the additional results are also similar to those obtained in the SMILES-to-SMILES experiments (**Figure 6**).

The latter point is extremely important with regard to the concept of the present study. First of all, as you pointed out, the true challenge for a model is not to learn lots of easy operations, but to learn difficult operations that remain until the end of learning. To change the point of view, we evaluated the percentage of correct answers for each character in order to determine which atom-binding rules were easy and which were difficult for the model to understand. As a result, we found a slow increase of the accuracy of "@" and "@@", the chiral tokens, compared with other tokens (**Supplementary Figure 5**). Notably, we found that 1) most of the wrong answers in stagnation are related to chirality (**Figure 4c**), and 2) the accuracy transition for chirality is slower than for other tokens with or without stagnation (**Supplementary Figures 6 and 7**). In other words, it was suggested that the most difficult operations for the model to learn are those related to chirality, and that the persistence of such misunderstanding manifests itself as stagnation in the learning curve. It should be noted that the accuracy increase of "\ " token, related to geometrical isomers, was also slow (**Supplementary Figure 5**). This suggests that the tokens associated with stereochemistry pose a general challenge for the Transformer model in terms of learning, while the influence of geometrical isomers remains relatively modest, owing to their infrequent occurrence (approximately 1/100th of the chiral tokens). The shift in viewpoints has elucidated this observation regarding the geometric isomers.

We added a discussion based on the results of the above points and the viewpoints we received from you. Consequently, stagnation is a condition under which difficult operations are likely to be

manifested. While the singularity of chirality persists unaltered, a broader discourse has become feasible, owing to the shift in viewpoint. Once again, we deeply appreciate you for your valuable input.

Modification

- ✓ Comparison of partial/perfect accuracy, Tanimoto similarity of molecular fingerprints and loss function of an InChI-to-SMILES model were shown (**Figure 6a and 6b**).
- ✓ Downstream task performance of InChI-to-SMILES model was studied and shown (**Supplementary Figures 10 and 11**).
- ✓ Character-wise accuracy of InChI-to-SMILES model with pre-LN structure was studied and shown (**Figure 6c**).
- ✓ Classification of mistakes of InChI-to-SMILES model according to chirality was studied and shown (**Figure 6d**).
- ✓ Transition of character-wise accuracy of random-to-canonical translation was added, and it was argued that the operations on chiral tokens are difficult for the model to recognize as structure-altering (**Supplementary Figure 5**).
- ✓ The parts related to this point were substantially revised according to the comments (page 14, line 358-372)

Comments 2: In the context of tokenized SMILES, what the machine appears to be learning are not the substructures, which are essentially a handful of atom types, but rather the connectivities between these atom types. This arguably forms the underlying reason why predicting the entire molecular structure necessitates a longer learning duration as compared to the partial structures. However, the paper does nowhere explicitly address this aspect of learning connectivities, an omission that potentially weakens the central argument.

Answer

We appreciate your insightful consideration and fully agree with you. We added the explanation of our results to the main manuscript that the essential task in SMILES translation is to reproduce connectivity of atom types, that the model in the early stages of learning can only reproduce easy connections which are observed as partial structures, and that relatively difficult connections which join these partial structures require more training for the model to understand.

Modification

- ✓ The above considerations regarding the understanding of partial and overall structures were added (page 10, line 242-246 and page 10, line 265-272).
- ✓ The wording in the related sentences were changed based on the above considerations.

Comments 3: The paper exhibits an exceedingly high level of abstraction, with an unclear transition from the sequential treatment of SMILES tokens to the substructural analysis performed on MACCS and ECFP features. The training is conducted through SMILES tokens, but the authors execute a substructural analysis at both ends using entirely different substructural algorithms. This approach makes the entire analysis dependent on how substructures are defined, which could limit the generalizability of the findings. The interpretation of such a fundamental task can be effectively conducted by contemplating on the Maximum Likelihood principle.

Answer

We apologize for insufficient description of the experiment comparing the input and output characteristics of MACCS and ECFP. This experiment was conducted primarily to increase interpretability to the broader scientific community. As shown in Figure 1 and as noted in your comments, the Transformer model easily captured the partial connectivity of the atomic types during the study. However, we believe that it is difficult to imagine "what kind of partial connectivity of

atomic types can be easily captured". Therefore, to translate this into more general language, we compared the progress of input-output matching at the substructure level defined in MACCS and ECFP. While these fingerprints deal with limited and predefined substructures as you pointed out, these are well known and employed in various scientific fields such as chemistry, pharmacology, and toxicology, which we believe helps various readers in *Nat Commun* to grasp what we found in this study. According to your comments, we have modified the manuscript as follows.

Modification

- ✓ It was written that as these 2 descriptors represent typical partial structures of molecules, their similarity between target and prediction can be an indicator of the comprehension of the model on partial structures of molecules (page 10, line 251-253)

Comments 4: The manuscript seems to be written with some oversight, containing several redundant sentences (e.g., lines 141-146) and typos (e.g., lines 79, 285).

Answer

We deeply apologize for our insufficient confirmation about the manuscript and appreciate your comment. We checked and modified the mentioned sentences and reviewed the entire text.

Modification

- ✓ The description about dimension-wise MACCS keys similarity was shortened (page 7, line 160-163).
- ✓ Typos in the manuscript were checked and modified.

Reviewer: 3

This manuscript tried to investigate the NLP model for understanding chemical structure. They employed the Transformer and investigated the status during the learning stage. Although they claimed the NLP model has difficulty for understanding the chirality, the current data is not enough to support this. The writing is hard to understand, and the logic is not very clear. More importantly, they didn't provide solution for this.

Major Comments 1: Although they defined two accuracies to evaluate the learning process, the transformer itself is a blackbox, and it may be determined by many factors instead of chirality. The authors need to provide a way to solve this problem for publication in this prestigious journal.

Answer

We appreciate your essential comment. As you mentioned, it is unclear how the Transformer model learns chemical structures, and in this study, we address to this very question by monitoring the relationship between learning progress and structural properties of input/output, including evaluation of the two accuracies. Specifically, we compared the reconstruction of fingerprints and confirmed that partial structures of molecules are learned earlier. We also compared the agreement for each dimension of MACCS keys, and clarified no partial structures indicated by MACCS keys are particularly difficult to understand. We examined the performance transition of the descriptors in predicting molecular properties, and found that the information required for property prediction is included in the descriptors from the early stage of learning, i.e., such information is understood by the model. Furthermore, we found that Transformer is not good at understanding chirality, and that pre-LN structure facilitates its comprehension. According to your comments, we have revised the manuscript to articulate that these studies are objected to clarify the black-box of Transformer as follows.

Modification

- ✓ In the introduction and conclusion, we stressed that the objective of this paper is to understand how the Transformer model understands chemical structures (page 3, line 34-38 and page 18, Line 473-474).

Major Comments 2: The authors didn't make comparison with SOTA deep learning methods for property prediction. The current embedding produced results comparable to or even worse than ECFP & CDDD.

Answer

We apologize for not comparing SOTA models with our descriptor about the downstream task performance without clear explanation. We had not conducted comparison with SOTA models because our research was not directly objected to improve the performance of a certain molecular property prediction task, but to understand the property of the model and the descriptor.

However, as you mentioned, it helps to understand the property of the model to compare the performance with the those of the current SOTA models, and we replicated one of such models and used as a baseline. We think the relatively low performance of our descriptor can be attributed to no additional improvement of the model for molecular property prediction, such as co-learning of molecular properties which is adopted in CDDD descriptor. Such improvements can be prospective future works. According to your comments, we have modified the manuscript as follows.

Modification

- ✓ One of the SOTA models in molecular property prediction, Uni-Mol¹ was replicated and the performance of it was shown as baseline models. (**Figure 2, Supplementary Figure 1, 2, 10, 11, 18, 19**). This model is the SOTA model in some of the MoleculeNet² tasks in September 29³.
- ✓ It was noted that no structural optimization for property prediction is tried in this study, which can be the reason for relatively low accuracy (page 12, line 300-302).

Major Comments 3: Case examples are desired to deep into the structure. Currently, only overall

statistics is not enough.

Answer

We apologize for our insufficient illustration to support our argument. About recognition of partial and overall structures of molecules, we showed 10 examples of molecules and the prediction of them by the model at an early step of training (step 4000), and demonstrated that only partial structures are correctly decoded (**Figure 1c**). Additionally, we increased the graphical examples of molecules which are correctly predicted except chirality in stagnation (**Figure 4b**).

Modification

- ✓ 10 valid examples of molecules in test set and their prediction at step 4000 were randomly sampled and shown (**Figure 1c**).
- ✓ The examples of molecules and their prediction at step 10,000 were increased (**Figure 4b**).

Major Comments 4: The writing needs improvement. The method section is not clear. For example, the dataset in the first processing step was 30 M, but what's the final number? It's also common to filter the original dataset and then sample the dataset. They also mentioned "conducted the key experiments with the dataset prepared by random sampling", then what's the details of the dataset. "amounted to about 150,000" needs an exact number. "aborted some of the experiments when accuracy reached..."

Answer

We deeply apologize for our insufficient explanation about the methods of the experiments. We clearly explained the numbers and methods you mentioned and checked the entire manuscript for vague expressions.

Modification

- ✓ The number of sampled molecules in the training set and steps in one epoch was examined and specified for both random and stratified sampling (page 5, line 100 and **Supplementary Note 1**, line 14).
- ✓ The detailed procedure of stratified sampling was explained (page 5, line 98-106).
- ✓ The number of aborted studies was stated for the experiments about different random seeds and iteration order (page 8, line 200-201).

Major Comments 5: The authors need to carefully proofread/clarify the manuscript. "To the best of knowledge", "the limitation of study"; "to confirm generality and confirmed similar results"; "we calculated perfect accuracy with the selected character being masked. This means we did not check whether the selected character was correctly predicted by the", etc.

Answer

We would like to express our deepest apologies for our poor English and sincere thanks for reading the manuscript. We modified wrong or irrelevant expression you mentioned and proofread the entire manuscript. Please note that we employed an English proofreading service by ONLINE English (<https://www.oleng.com.au/>).

Modification

- ✓ Spelling errors in the manuscript were checked and modified.
- ✓ Ambiguous expressions in the manuscript were checked and clarified.

References

1. Zhou, G. *et al.* UNI-MOL: A UNIVERSAL 3D MOLECULAR REPRESENTATION LEARNING FRAMEWORK. Preprint at <https://chemrxiv.org/engage/chemrxiv/article-details/6402990d37e01856dc1d1581> (2023).
2. Wu, Z. *et al.* MoleculeNet: A benchmark for molecular machine learning. *Chem Sci* **9**, 513–530 (2018).
3. Molecular Property Prediction. <https://paperswithcode.com/task/molecular-property-prediction>.

REVIEWERS' COMMENTS

Reviewer #1 (Remarks to the Author):

The authors addressed the raised questions and improved the manuscript accordingly.

Reviewer #3 (Remarks to the Author):

I confirmed that the quality of the manuscript has been improved according to my previous comments.

Graphical and numerical data added to the manuscript certainly reinforced the conclusion.

However, I would like to raise a concern regarding the specific focus of the topic; it may be perceived as too specialized for the readership of Nature Communications.

Furthermore, in the revised manuscript, specifically between lines 358-372, the authors have clearly concluded that learning chirality poses a formidable challenge for models, resulting in stagnation.

This observation aligns with a prior finding wherein they quantified the effect to be approximately 20% using the ECFP4 fingerprint.

The authors are encouraged to refer to the following report: [<https://jcheminf.biomedcentral.com/articles/10.1186/s13321-023-00693-0>].

Reviewer #4 (Remarks to the Author):

I have no further comments.

Point-by-point responses to reviewers' comments

Reviewer Comments to Author:

Reviewer: 1

The authors addressed the raised questions and improved the manuscript accordingly.

Answer

We wish to express our deep appreciation to the Reviewer for reviewing our manuscript.

Reviewer: 3

I confirmed that the quality of the manuscript has been improved according to my previous comments. Graphical and numerical data added to the manuscript certainly reinforced the conclusion.

However, I would like to raise a concern regarding the specific focus of the topic; it may be perceived as too specialized for the readership of Nature Communications.

Furthermore, in the revised manuscript, specifically between lines 358-372, the authors have clearly concluded that learning chirality poses a formidable challenge for models, resulting in stagnation.

This observation aligns with a prior finding wherein they quantified the effect to be approximately 20% using the ECFP4 fingerprint.

The authors are encouraged to refer to the following report:[<https://jcheminf.biomedcentral.com/articles/10.1186/s13321-023-00693-0>].

Answer

We appreciate your valuable comments. Based on your helpful feedback, we have enhanced the discussion by adding an appropriate reference. We have modified the manuscript as follows:

Modifications

- ✓ “The true challenge for a chemical language model lies not in mastering the numerous elementary atom-bonding rules, but rather in comprehending the difficult rules that persist even after the basics have been acquired, which is consistent with Ucak’s work regarding the reconstruction of molecular representations from fingerprints”. (page 7, lines 183-186)

Reviewer: 4

I have no further comments.

Answer

We wish to express our deep appreciation to the Reviewer for reviewing our manuscript.